# Aroma Characterization of Roasted Meat and Meat Substitutes Using Gas Chromatography–Mass Spectrometry with Simultaneous Selective Detection and a Dedicated Software Tool, AromaMS

**DOI:** 10.3390/molecules28093973

**Published:** 2023-05-08

**Authors:** Nitzan Tzanani, Ariel Hindi, Dana Marder

**Affiliations:** 1Israel Institute for Biological Research, Ness Ziona 7410001, Israel; mambi@iibr.gov.il; 2Institute of Chemistry, The Hebrew University, Edmond Safra Campus, Givat Ram, Jerusalem 91904, Israel

**Keywords:** gas chromatography–mass spectrometry, selective detectors, simultaneous detection, aroma analysis, plant-based meat substitute, software tool

## Abstract

The development of healthier and more sustainable food products, such as plant-based meat substitutes (PBMSs), have received significant interest in recent years. A thorough understanding of the aroma composition can support efforts to improve the sensory properties of PBMS products and promote their consumer acceptability. Here, we developed an integrated hardware and software approach for aroma analysis of roasted food based on simultaneous analysis with three complementary detectors. Following the standard procedure of aroma headspace sampling and separation using solid-phase microextraction-gas chromatography, the column flow was split into three channels for the following detectors for the selective detection of nitrogen and sulfur (N/S)-containing compounds: an electron ionization-mass spectrometry for identification through a library search, a nitrogen-phosphorous detector, and a flame-photometric detector (FPD)/pulsed-FPD. Integration of results from the different types of detectors was achieved using a software tool, called AromaMS, developed in-house for data processing. As stipulated by the user, AromaMS performed either non-targeted screening for all volatile organic compounds (VOCs) or selective screening for N/S-containing VOCs that play a major role in the aroma experience. User-defined parameters for library matching and the retention index were applied to further eliminate false identifications. This new approach was successfully applied for comparative analysis of roasted meat and PBMS samples.

## 1. Introduction

The use of fire in food preparation represents a milestone in human history, allowing the roasting of meat after the conclusion of a successful hunt. The aroma of roasted meat is generated through the heat-induced Maillard reaction, a chemical reaction that occurs between amino acids and reducing sugars, combined with the breakdown of fats. A mixture of a variety of volatile organic compounds (VOCs) contributes to the overall aroma of roasted meat [1,2], including but not limited to pyrazines, pyrroles, thiols, sulfides, thiazoles, aldehydes, furans, esters, and lactones. The synergistic effect of these numerous compounds produces the desired aroma and flavor of roasted meat, and the aroma composition is highly dependent on both the exact ingredients of the raw food and the heating profile of the roasting procedure.

Although meat is the main dish in the cuisines of most countries, over the last decade, a growing interest in plant-based meat substitutes (PBMSs) has emerged. PBMS products are attractive for several reasons, with the top three being sustainability, animal welfare, and health benefits. Meat production is a major contributor to greenhouse gas emissions, resource consumption, and waste production, while PBMSs have a much lower environmental impact. PBMSs are also an ethical alternative to traditional meat for those concerned about animal welfare and can be a good option for those seeking to follow a vegetarian or vegan diet owing to health concerns. In the field of PBMSs, the challenge for research and development teams is to achieve formulations that successfully create the experience of meat dishes, particularly with regard to the aroma and flavor, while incorporating ingredients of plant origin. Currently, the common approach for aroma evaluation in the food industry relies on professional sensory teams, but the incorporation of instrument-based analysis can provide the advantages of high throughput and reproducibility. While fingerprint-based techniques have been applied to aroma characterization, including that for heat-treated meat [3,4,5], a more informative approach is to perform a detailed chemical analysis [6]. Instrumental analysis is highly sensitive and specific, making it capable of pinpointing the exact differentiating compounds between samples. The challenge for the analytical chemist in supporting PBMS research and development is to detect, identify, and quantify the VOCs above PBMS dishes so they can be compared with those of the corresponding meat dish. This challenge is exacerbated by the high complexity of the aroma mixture, which is composed of hundreds of VOCs, among which are minor constituents that can have high odor activity values (OAVs).

In accordance with the complex chemical composition of food aromas, the method of choice for the analysis of VOCs is the gas chromatography–mass spectrometry (GC–MS) tandem technique. Headspace sampling with solid-phase microextraction (SPME) is often coupled to GC–MS for sample introduction, including for aroma analysis of meat products [7,8] and investigation of how meat aroma changes with different preparation steps [9,10]. Various statistical methods, including principal component analysis and partial least squares-discriminant analysis (PLS-DA), have been employed for chemometrics in SPME-GC–MS data processing in the search for significant markers [11,12,13,14]. GC offers high separation power for VOCs with good correlation between the retention time (RT) and volatility. GC can be coupled with a variety of detectors, each with their own capabilities and limitations. Electron ionization (EI)-MS provides semiquantitative detection of VOCs, with identification achieved by a library search, but without preferences for significant odorants with high OAVs. While an olfactory detector coupled to GC provides a sensory assessment of food aroma components, its coupling to GC–MS is of specific value for the identification of compounds with high OAVs [15,16,17,18,19]. Selective detectors provide sensitive detection and quantification of heteroatoms, even in the presence of a heavy matrix. Sulfur is known to induce high OAVs; an atomic emission detector operating in a sulfur-selective mode was used together with GC–MS and GC-olfactometry to identify sulfur-containing odorants [20].

Amirav [21] and Dagan [22] demonstrated that coupling GC–MS with selective detectors can give a synergistic advantage for the orthogonal information. The correlated chromatograms from selective detectors and EI-MS support the sensitive and reliable identification of pesticides that contain heteroatoms, such as phosphorus and sulfur. Although simultaneous detection has been demonstrated for the detection and identification of pesticides in agricultural products, it has the potential to provide valuable information on organo-nitrogen and organo-sulfur VOCs that play a central role in food aroma. To the best of our knowledge, the simultaneous detection approach has not been widely accepted in analytical laboratories, including those in the food industry.

Here, we present the application of simultaneous analysis by operating selective detectors parallel to MS to support the aroma characterization of roasted meat and PBMSs. The GC column flow was split into three channels for the following detectors: MS, a nitrogen-phosphorous detector (NPD), and a flame-photometric detector (FPD(S))/pulsed-FPD (PFPD) (herein (P)FPD(S)) for the identification of N-containing and S-containing compounds (herein N/S compounds). The information from the selective detectors, synchronized with MS, enabled N/S compounds to be highlighted among the hundreds of other VOCs detected by MS. The data processing of the multidimensional output of the GC–MS/NPD/(P)FPD(S) system is labor intensive and involves an MS library search with the integration of results in view of selective chromatograms, necessitating automation. An in-house software tool called AromaMS was developed to perform all the required tasks including total ion chromatogram (TIC) deconvolution (using AMDIS), target and non-target MS library searching, filtration by the retention index (RI), correlation of chromatograms, and verification of tentative identifications using selective detection results.

## 2. Results

### 2.1. GC–MS/NPD/(P)FPD(S) Configuration

The system configuration is based on the GC–MS/(P)FPD(S) system described by Amirav [21]. The configuration uses an injection port connected to a capillary column with splitting of the carrier gas flow between EI-MS and the selective detectors to obtain two-dimensional information regarding the compounds in the sample. In our system, to maximize its usefulness, an NPD detector and a multimode inlet (MMI) were also incorporated. As PFPD and FPD(S) detectors are similarly beneficial for the analysis of roasted food, herein we refer to the system as GC–MS/NPD/(P)FPD(S). This configuration of interconnecting two inlets and three detectors to a single operating unit is depicted in Figure 1. Each inlet is connected separately to a capillary column, and the gas flows are combined using two ports of a three-way splitter with a make-up gas. Part of the combined flow goes through a third port to the MS through a 100 μm diameter short column where the flow rate is controlled by the pressure difference between the splitter and the vacuum in the MS. The rest of the flow is evenly split between the selective detectors using a three-port splitter. Both splitters have a highly inert surface and minimal dead volume to retain good chromatographic performance.

While the (P)FPD(S) detects the existence of S-containing compounds, the incorporation of the NPD adds an orthogonal dimension to the analysis results: the selective detection of N-containing compounds, such as pyrazines, pyrroles, and amines, that elute among the abundant hydrocarbons and O-containing compounds. The sensitivity and dynamic range of PFPD(S), FPD(S) and NPD(N) are comparable to the sensitivity of the MS system; hence all N/S compounds amenable to MS identification can be detected in the selective chromatograms. To enable simultaneous detection with the three detectors without peak broadening, the total gas flow was maintained at 0.8–1 mL/min per detector. While the gas flow in the separation column was kept at 2 mL/min, to balance the run time vs. separation, the additional flow from the secondary column and make-up gas brought the total flow into the desired range of 2.5–3 mL/min. For the current study, SPME injections were performed at the MMI, while an autosampler was installed at the back inlet for injections of standards and calibration mixtures at the beginning of the day. Although the incorporation of two inlets is not essential for simultaneous detection, our experience has shown it to be useful for the implementation of complementary methods. Both inlets are suitable for sample introduction, and prior to each analysis, the analyst can choose which inlet to use, by simply loading the suitable method. For example, in previous works the front inlet was used for SPME injection of gases and volatiles on a PLOT column or for the large-volume injection of extracts.

The synchronized results from the three detectors are automatically stored in the same data folder, and can be viewed together with the MassHunter Qual browser, which also supports time-scale shift of the chromatograms for RT alignment. As discussed in detail by Amirav [21], there is only a slight restriction between the splitters and the atmospheric pressure in the detectors; therefore, the pressure in the connectors is almost constant and only slightly above atmospheric pressure, keeping the gas flow to the MS system and detectors stable during the chromatographic run. If the gas flow in the columns is kept constant, the RT lags of the selective detectors relative to the MS are stable and limited mainly by the uncertainty in the determination of the peak apex. The RT difference between the MS system and the NPD and FPD was 0.4 s, with a narrow tolerance window of 0.5 s, while the RT difference between the MS system and PFPD and was 3.5 s due to the electronic delay of the PFPD, with a similar window of 0.5 s. The accuracy of the RT alignment between the detectors is useful to detect N/S compounds in the TIC, extract their spectra even when they are only partially separated from adjacent peaks, and submit them to a library search. However, this approach toward data processing is highly time consuming even for a few compounds, and the roasted meat aroma contains dozens of N/S-containing compounds. For an exhaustive processing of the data from simultaneous detectors, we developed an in-house software tool called AromaMS.

### 2.2. AromaMS: Data Processing Software Tool

AromaMS is designed to receive the analysis results from a GC–MS/NPD/(P)FPD(S) system as input and export a list of identified compounds, as described in the workflow in Figure 2. AromaMS has two main modes of operation: 1. Targeted, searching the library of the user, and 2. Nontargeted, searching a public library, such as NIST mainlib. For each mode of operation, there is an option to verify the search results with the results of the selective detectors. Non-targeted screening is at the center of our current research, mainly with the confirmation of selective detectors. After the sample analysis is completed, the raw-data file is analyzed in several steps: 1. Data file is loaded to MSD Chemstation for integration of the peaks in selective chromatograms, followed by the “percent report” command that exports the integration results to a file named “rteres.txt.”; 2. AromaMS runs AMDIS that deconvolutes MS data and saves the output spectra in a text file (ELU format); 3. AromaMS then sends each spectra, one by one, to search the NIST EI library and reads the results, filtering the results by RI values on a non-polar column; 4. if “detectors verification” is “on”, AromaMS reads the report of the selective detectors with the integration results, which includes the peaks of the three channels based on the user-defined time lags and tolerance; 5. AromaMS prepares the summary report that includes either identifications of all compounds in the sample or only compounds containing heteroatoms with confirmation from the selective detectors, such as nitrogen with the NPD and sulfur with the FPD(S)/PFPD.

The final reports are exported as *.docx files (one for each operation mode) with a table containing the detailed record for each identified compound. The details available for each compound include the RT, name, CAS number, NIST scores (Match, Reverse match, and Probability), MS peak attributes (S/N, area), and selective detector indications. Examples of reports for the targeted and non-targeted modes are given in the Appendix A. AromaMS is also capable of additional actions, such as RI calibration of AMDIS and semi-quantitation of identified compounds against reference standards, but these features are secondary and are not described herein.

AromaMS has a graphical user interface for parameterization, as shown in the Appendix A. The main screen (Appendix A) is used by the operator to set the following user-defined parameters: raw-data file name and path; mode of screening, targeted and/or non-targeted; with/without selective detector verification; types of active selective detectors with their respective RT difference; with/without RI filtering; and with/without quantitation. The administrator screen (Appendix A) is used to enter the more fundamental settings for AromaMS, including the NIST path, AMDIS operational parameters, and list of selective detectors. When using AromaMS for data processing, entering the full name of the data file allows AromaMS to find the files that contain the MS data and selective detector integration results, AMDIS TIC deconvolution results, and NIST library search results. Parameterization of a data processing software gives a trade-off between sensitivity and reliability, since trying to identify as many compounds as possible leads to a large number of false-positive results. With the parameterization of AromaMS, we sought to achieve balanced performance. The TIC deconvolution parameters were set to medium sensitivity and shape requirement, while the use of a capillary column achieves high resolution. The match score was set to 700, and the RI tolerance was set to 100 based on our experience using the NIST mainlib for non-target screening. The RT alignment tolerance of the selective detectors was set to 0.02 min to accommodate cases of partial separation that can hamper the measurement of the peak apex.

AromaMS is freely available for R&D and evaluation purposes, with full support for installation and operation, upon contacting the authors.

### 2.3. Aroma Characterization of Roasted Meat

The headspace of roasted meat is a mixture of a large number of VOCs; hence GC–MS analysis yields a highly complex TIC that contains hundreds of compounds, ranging in intensity over several orders of magnitude. Figure 3 demonstrates this complexity with an example SPME-GC–MS/NPD/FPD(S) analysis of a roasted meat sample, where the TIC in trace A comprises numerous peaks that are partially separated at best. Identification of all compounds in the TIC is an option, using either a user-built library or public library, such as the NIST library. However, small potentially overlooked peaks may come from compounds with high OAVs that may make a significant contribution to aroma properties. Traces B and C in Figure 3 show the respective chromatograms obtained with the NPD and FPD(S) results of this analysis synchronized with the TIC in trace A. As the chromatograms of selective detectors are not as complex as the TIC, each peak can be examined and identification of the compound can be attempted knowing that the compound is likely to be a significant component. The use of the selective detectors as a guide to search for N/S-containing compounds in the TIC, as described above, was applied to six prominent compounds, tentatively identified as aminoethanol, sulfide, and several pyrazines. The library search results are shown in the top panel of Figure 3.

The ability to achieve simultaneous analysis using the GC–MS/NPD/(P)FPD(S) system to discern the closely eluting compounds is demonstrated in Figure 4, which zooms-in on a 0.5 min time interval of the TIC. In the chromatograms of the selective detectors, five N-containing compounds and one S-containing compound were detected, most of which were barely observable in the TIC. Once we learned their exact RTs and heteroatom contents from the selective detectors, the respective spectra were searched and tentatively identified. One such example is methyl pyrrole (at 2.271 min), which was unresolved from the previous and considerably more abundant methyl ester (at 2.226 min). Pyrrole and dimethyl disulfide peaks were only partially resolved, while methylpyrrole and methylpiperidine were hardly observable in the TIC. These four compounds elute in the narrow time period of 2.27–2.4 min (8 s) and are certainly significant to the aroma, although they are present in low abundances. To enable a thorough characterization of the aroma by examining the entire data file, we continued data processing using AromaMS in the non-targeted screening mode. The results, both with and without selective detector verification, are included in the Appendix A (Appendix A, respectively). While non-targeted screening yielded a list of 150 compounds, comparison with the selective detector results reduced the list to 62 compounds.

GC–MS/NPD/FPD(S) was similarly applied to the analysis of the headspace above roasted spoiled meat, and the results are shown in Figure 5. The chromatograms indicate that the aroma of spoiled meat after roasting is characterized by four S-containing compounds that are known to induce a pungent odor. These compounds were hardly noticeable in the TIC, but by using the RTs of the FPD(S) peaks, we could extract the corresponding peaks in the TIC and use them for a library search. The compound that eluted first and was hardly retained was possibly identified as methanethiol, while the other compounds were tentatively identified as dimethyl disulfide, dimethyl sulfone, and dimethyl trisulfide. Not only were the MS signals of these compounds weak compared to those of the other VOCs, but three of the compounds were only partially separated from more abundant adjacent peaks. Dimethyl sulfone was only a shoulder on the peaks of furanone and butyrolactone, while dimethyl trisulfide was on the slope of an interfering siloxane peak. For this sample, the full AromaMS results for non-targeted screening with selective detector verification are shown in Appendix A.

### 2.4. Comparative Analysis: Meat vs. PBMSs

Simultaneous analysis using MS and selective detectors has additional value in the comparative analysis of samples, highlighting the differences in N/S compounds. This capability is demonstrated in Figure 6 when this method was applied to the aroma characterization of roasted meat samples and two PBMSs purchased at a food store. The analysis was performed using GC–MS/NPD/PFPD, displaying three overlaid chromatograms for each detector. As is evident from trace A, the TICs differ from each other, but clear conclusions regarding the aroma experience cannot be drawn from the TICs. From trace B, with the overlaid NPD chromatograms, interpretation is more straightforward. Certain compounds are more abundant in the meat sample, especially the tentatively identified trimethylamine, dimethylpyrazine, and trimethylpyrazine. On the other hand, dimethyl aminoethanol is more abundant in PBMS 2. Examination of the overlaid PFPD chromatograms in trace C reveals that methanethiol is less abundant than in the meat sample. In PBMS 1, an early eluting highly volatile unidentified S compound and thiirane are identified as being relatively abundant, while in PBMS 2, dimethyl sulfide and dimethyl disulfide are relatively abundant. The three data files were processed using AromaMS via non-targeted screening with N/S-containing compound verification, and the detailed results are given in the Appendix A.

## 3. Discussion

Food aromas, such as those of roasted meat and PBMS products, are complex mixtures containing hundreds of compounds, many of which are O-containing compounds. GC–MS analysis is an efficient instrument-based method for the detection of VOCs, offering the advantages of sensitivity, reproducibility, and high-throughput analysis. However, the thorough identification of VOCs using GC–MS and non-targeted screening is labor intensive, time consuming, and error prone. Therefore, a comparative analysis of two food products to assess similarities and differences is particularly challenging. The uniform response of GC–MS to organic compounds means that prominent compounds in the TIC do not necessarily make significant contributions to the sensory properties.

The combination of selective detectors to analyze the eluents in parallel with MS provides orthogonal dimensions of information, highlighting compounds containing heteroatoms known for their high OAVs. The NPD detects compounds such as pyrazines, pyrroles, and pyridines, and the FPD(S)/PFPD detects compounds such as thiols and sulfides, while both detectors detect thiazoles, which are well-known to have a significant contribution to the aroma of roasted meat and PBMSs. There are three advantages to the simultaneous detection approach: support/discard library search results, detection of low-concentration compounds that would otherwise be missed, and direct comparison between samples even without the detection of all compounds. A GC–MS configuration with two inlets connected to two separation columns with gas flows combined and then split to three detectors has been shown to be robust for operation and accurate for RT correlation. This type of system, called GC–MS/NPD/(P)FPD(S), provides multidimensional information about the samples, and therefore requires appropriate software to extract all the meaningful information regarding aroma components. AromaMS was specifically developed to meet the challenges associated with the required extensive processing of GC–MS/NPD/(P)FPD(S) results, library search for all compounds in the non-targeted state, data extraction on N/S compounds, and correlation of the two data sets, to achieve both comprehensive identification of VOCs while highlighting those that contain N/S heteroatoms. This combination of hardware and software enables the informative analysis of dozens of samples per day to support the optimization of the formulation and preparation method of food products. It should be emphasized, however, that AromaMS was designed to streamline the multi-detector data processing task and cannot circumvent the inherent limitations of insufficient separation in spectra, false identifications, and missed identifications. Analysis in nontarget mode can provide only tentative identification of isomers, such as dimethylpyrazines, while unambiguous level-1 identifications require a follow-up injection of standards and analysis in target mode.

## 4. Materials and Methods

### 4.1. Reagents and Materials

Minced meat and PBMS of two brands (herein PBMS 1 and PBMS2) were purchased in the local food store. The meat was minced beef, while PBMS samples were plant-based burger substitutes.

### 4.2. Instrumentation

#### 4.2.1. Spme

StableFlex™ SPME fiber, with 65 μm PDMS-DVB coating was used for sampling, installed in SPME Fiber Holder (P/N 57326-U and 57330-U, respectively, by Merck, Rahway, NJ, USA).

#### 4.2.2. GC–MS/NPD/(P)FPD(S) System

Two similar configurations of GC–MS/NPD/(P)FPD(S) systems were used in this research. Both systems comprised 7890 GC equipped with two injection ports (MMI and S/SL), 5975C MSD in electron impact ionization mode with extractor source, NPD with Blos type bead and either FPD(S) (all by Agilent, Santa Clara, CA, USA) or PFPD (by OI) detectors. Separation columns for both inlets were capillary 15 m × 0.25 mm × 1 μm DB5-UI (P/N 122-5513UI by Agilent), connected to Helium make-up gas supply using three-way splitter with make-up gas (P/N G2855-61500 by Agilent) and then split to the MS and the selective detectors using a Silflow 3-port splitter (P/N 123,722 by Trajan, Ringwood, Australia).

GC–MS systems were controlled using MassHunter GC–MS acquisition version B.07.05, data processing was carried out using MassHunter Qualitative analysis version B.07.00 and MSD Chemstation F.01.03.2357 (by Agilent). NIST mainlib version 2020 was used for library search with the attached AMDIS version 2.73 and for TIC deconvolution (by NIST).

### 4.3. Methods

#### 4.3.1. Sample Preparation

Meat and PBMS samples were kept at 4 °C for less than a day before analysis. Spoiled meat was kept at room temperature for almost 48 h before analysis.

Roasting procedure: 0.5 g of sample was put in glass vial and covered with aluminum foil. The vial was placed in a GC oven, heated for 20 min at 180 °C and was left to cool down and equilibrate for 10 min.

#### 4.3.2. SPME Sampling

SPME fiber was inserted into the vial through the aluminum foil and left there for 5 min before taken for analysis.

#### 4.3.3. GC–MS Method

Inlet temperature was set to 250 °C, with vent time of 0.5 min. Helium was used as the carrier gas, flow rate in separation column was set to 2 mL/min with additional 0.5 mL/min in the second column and 2.5 psi make-up gas pressure. Oven temperature program began at 90 °C for 0.5 min, raised by 25 °C/min up to 315 °C and held for 0.5 min. temperatures and flow rates of the detectors were NPD heated to 300 °C, 3 mL/min H_2_, 120 mL/min air, 10 mL/min makeup N_2_; FPD(S) heated to 200 °C, 75 mL/min H_2_, 100 mL/min air, 60 mL/min makeup N_2_; and PFPD heated to 300 °C, 11 mL/min H_2_, 13 mL/min air, 14 mL/min makeup air.

### 4.4. Data Processing

AromaMS was developed in-house, written in Java (version 8).

#### 4.4.1. TIC Deconvolution

Following were the TIC deconvolution parameters using AMDIS: Component width = 10, Adjacent peak subtraction = 1, Resolution = High, Sensitivity = Medium, Shape requirements = Medium, and Use RI calibration = On.

#### 4.4.2. Library Search and Data Integration

NIST libraries search results were filtered using AromaMS with the following thresholds: ‘ΔRI < 100 & Match > 700’ or ‘Match > 800 & Probability > 20’, as not all records returned RI value. Correlation parameters of TIC and selective detectors chromatograms were set to ‘RT_NPD_—RT_MS_ < 0.01 min ± 0.02 min’, ‘RT_FPD_—RT_MS_ < 0.01 min ± 0.02 min’ and ‘RT_PFPD_—RT_MS_ < 0.05 min ± 0.02 min’.

## 5. Conclusions

GC–MS/NPD/(P)FPD(S) analysis is a new approach that produces information-rich results, allowing the efficient characterization of N/S-containing compounds in a complex mixture. This system is supported using AromaMS, a software tool that provides both thorough library search identification of VOCs and selective identification of N/S-containing compounds. The combination of GC–MS/NPD/(P)FPD(S) with AromaMS can greatly support the development of PBMSs by exploring the similarities and differences between their aroma and that of real meat, hence promoting the public acceptance of these products.

## Figures and Tables

**Figure 1 molecules-28-03973-f001:**
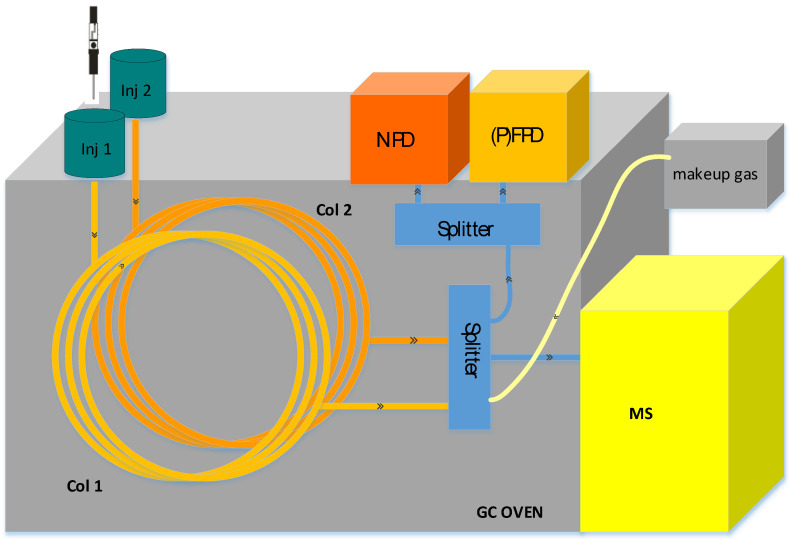
Configuration of the GC–MS/NPD/PFPD system. Two injection ports are connected to capillary columns, which are connected to a make-up gas supply with a four-way splitter. Gas flow is split to MS and two selective detectors using an additional three-way splitter.

**Figure 2 molecules-28-03973-f002:**
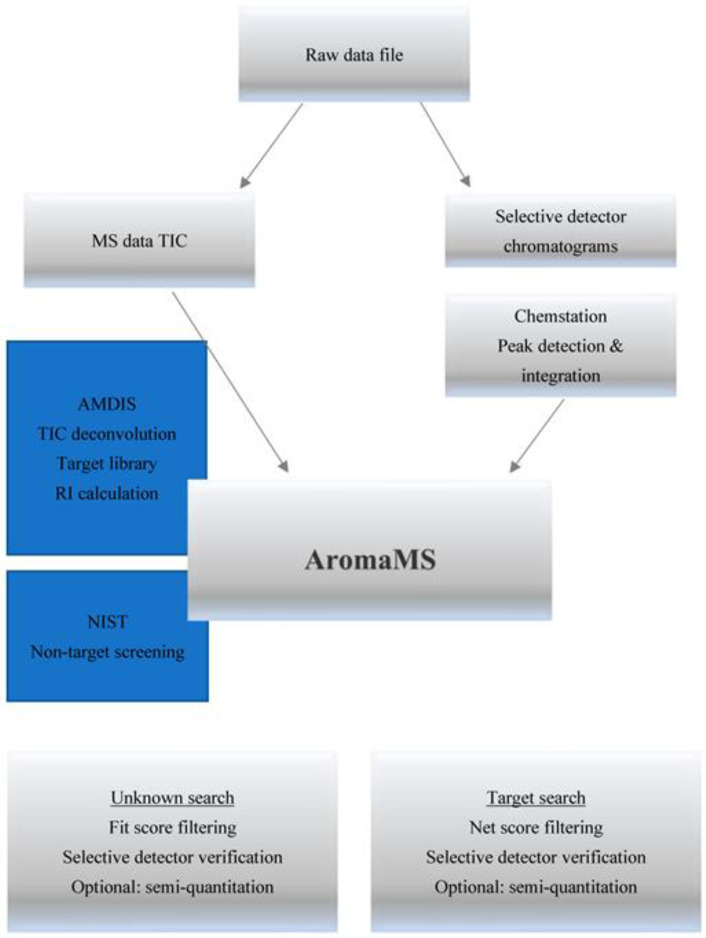
Workflow of AromaMS software tool, following TIC deconvolution with AMDIS and identifications in with or without verification with selective detection for nitrogen and sulfur.

**Figure 3 molecules-28-03973-f003:**
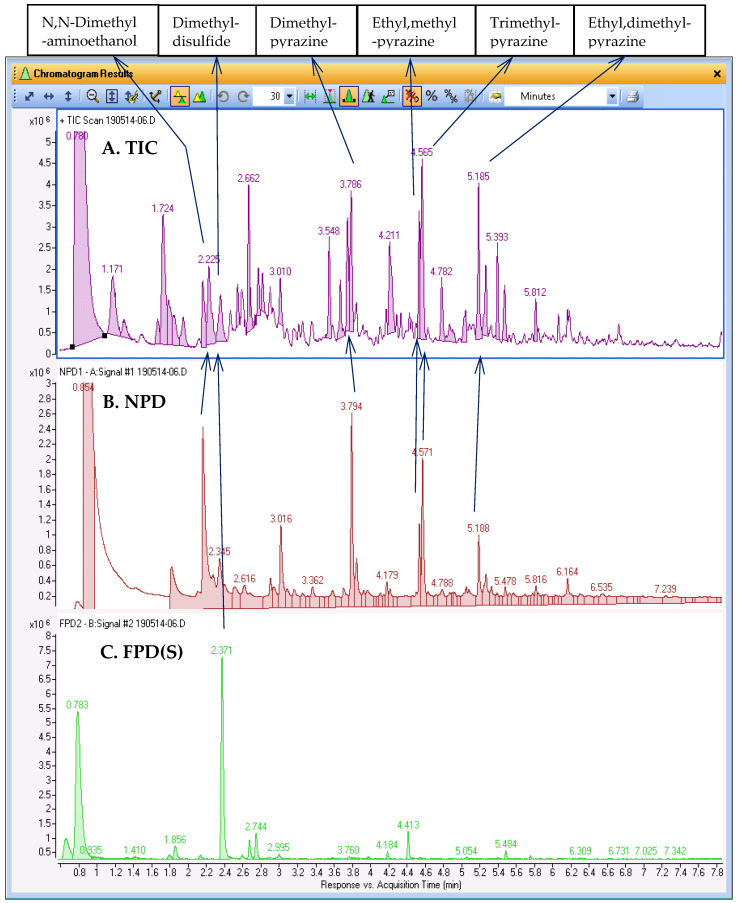
Analysis results of roasted meat aroma, using SPME-GC–MS/NPD/FPD(S) system. Trace (**A**) shows the TIC from the MS, trace (**B**) and trace (**C**) show the NPD and FPD(S) chromatograms, respectively. The selective identification of several prominent compounds that contain N or S atom is also presented. Compounds were tentatively identified based on search in NIST mainlib.

**Figure 4 molecules-28-03973-f004:**
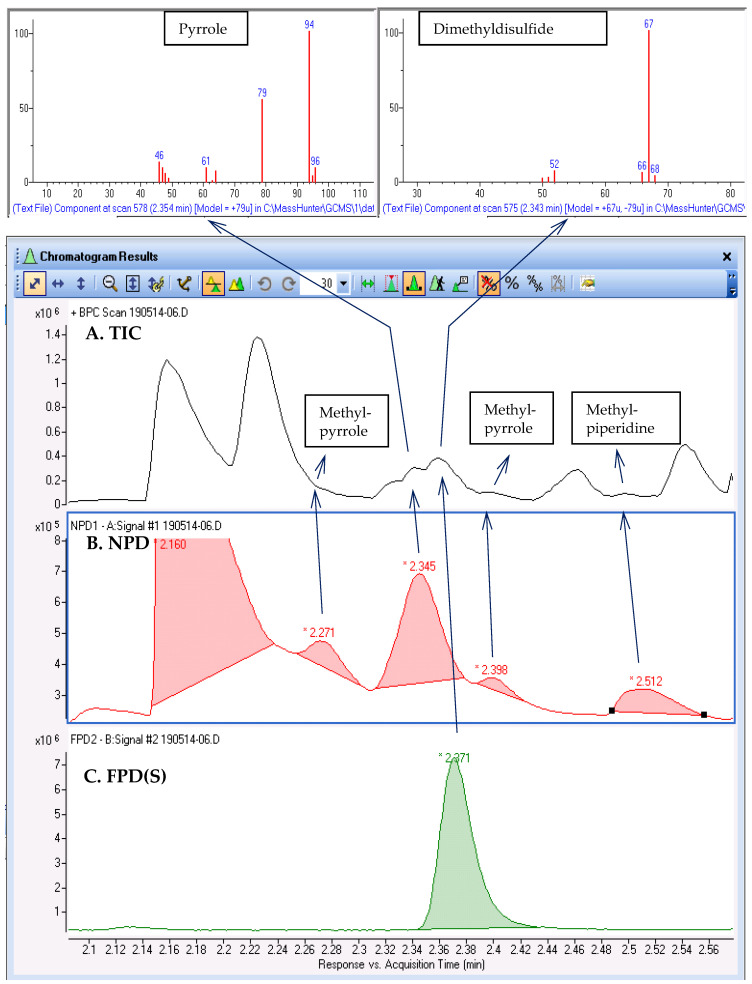
Selective detectors support for the identification of weak and unresolved compounds that contains N or S atoms. Compounds were tentatively identified based on search in NIST mainlib.

**Figure 5 molecules-28-03973-f005:**
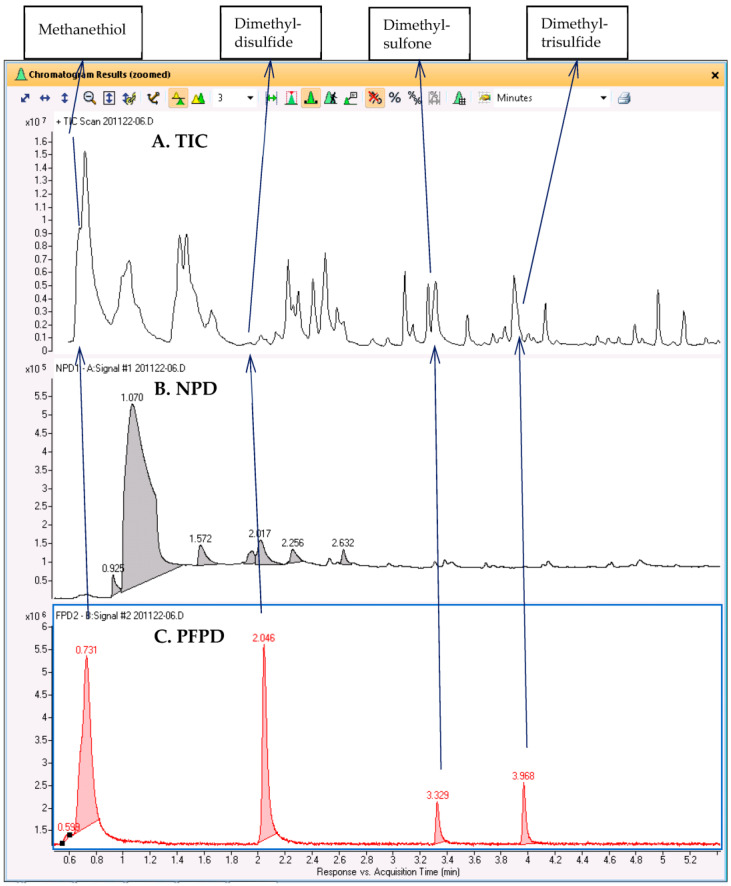
Analysis of roasted spoiled meat (180 °C for 20 min). (**A**) TIC; (**B**) chromatogram NPD chromatogram; (**C**) FPD(S) chromatogram. Compounds were tentatively identified based on search in NIST mainlib.

**Figure 6 molecules-28-03973-f006:**
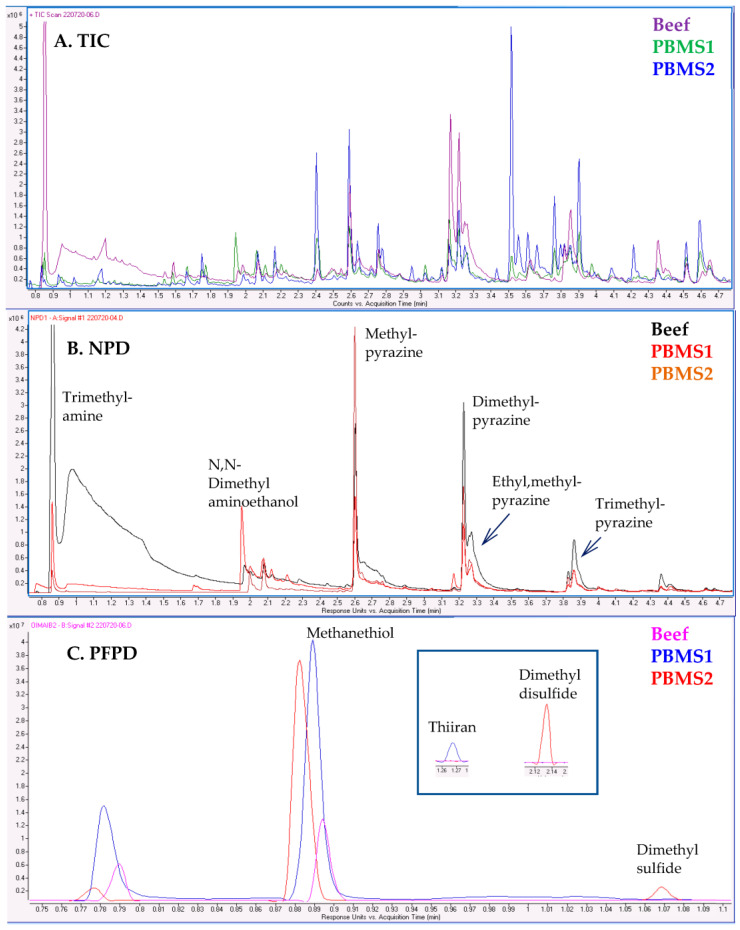
Comparative analysis of roasted meat and two PBMS, using SPME-GC–MS/NPD/PFPD analysis. (**A**) Overlaid TICs; (**B**) Overlaid NPD chromatograms; (**C**) Overlaid PFPD chromatograms, with zoom-in on the early part of chromatogram and inset with later eluting peaks. Compounds were tentatively identified based on search in NIST mainlib.

## Data Availability

Data are contained within the article.

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
