# Peer review of "Aroma Characterization of Roasted Meat and Meat Substitutes Using Gas Chromatography–Mass Spectrometry with Simultaneous Selective Detection and a Dedicated Software Tool, AromaMS"

_molecules, 2023, doi:10.3390/molecules28093973_

Round 1

Reviewer 1 Report

Dear Team,

I have been able to read and review an interesting work on analytical chemistry with relevant and promising results that could lead to a new, precise and faster technique for food analysis. The use of new food sources, such as those based on plants, could lead to new development alternatives and sources.

22 references are cited between 1998 (one) and 2019, mostly from 2019 onwards, which speaks of the novelty of the technique presented.

Some details to correct are:

Line 105, the author Amirav is cited, without reference (Nr 21) and says Error Bookmark.

Lines 107-108, redundancy in the wording... I propose changing to : in our system, to maximize its usefulness...

Line 139 describe how the analysis contributes or is modified, with what they describe as "optional manual corrections". It is warned as a contradiction in an automated system.

Lines 147-148, what difference is the acronym MS from MSD (Mass spectrum detector?) Throughout the document, it is the only time it is written like this and not as MS

Line 167. What happens with the software developed in-house, if for some reason it is necessary to carry out maintenance of the capillary column, cutting sections close to the injection port? Retention index (RI) is modified? Has this change been contemplated in the developed software?

Author Response

Dear reviewer,

Please find attached our revised manuscript molecules-2317462, “Aroma characterization of roasted meat and meat substitute using gas-chromatography mass-spectrometry (GC-MS) with simultaneous selective detection and the dedicated software tool AromaMS”, submitted for publication in Molecules. We were happy to learn that the topic is of interest to the scientific community.

In response to comments that were given to us, we have submitted the manuscript for English language editing by Editage Author service to improve its level of English writing.

We hope that you will now find the attached revised manuscript suitable for publication in Molecules.

Sincerely,

Nitzan Tzanani, PhD

Analytical Chemistry Department,

Israel Institute for Biological Research (IIBR)

Reviewer 2 Report

The authors developed an integrated system combining MS detector, FPD and NPD for the detection of aroma compounds presented in complex food samples. To facilitate the qualitative process, this paper also presents a home-build software package for the identification of aroma compounds. In general, this paper is well presented, and the results are interesting. 

Suggestions:

1. Figure 3-6 may present extracted ion chromatograms of detected compounds for the comparison between TIC and selective detector chromatograms.

2. AS a open access paper, the authors are encouraged to share the software package to repeat their work by other researchers.

3. Figure 1 is not quite clear for how to inject two samples to two separate inlets simultaneously.  

4. In Figure 1, What is the rationale of using two identical columns and then combine the gas flow after worth? instead of just using col 1 for mass spectrometer and col 2 for the other two detectors?

Author Response

(The authors gave the same response as above.)

Reviewer 3 Report

Aroma characterization of roasted meat and meat substitute using gas-chromatography mass-spectrometry (GC-MS) with simultaneous selective detection and the dedicated software tool  AromaMS

Nitzan Tzanani, Ariel Hindi and Dana Marder

Proper characterization of chemicals found in food samples is a very important endeavor. Study has undertaken such a challenge. The results are suitable for publication in “molecules.”  The authors have designed a program to streamline processing data from several detectors.  However, they accept that this protocol cannot circumvent the inherent limitations of spectral data, insufficient chromatographic separation, false and missed identifications.  I wish the authors evaluated the robustness of this method with the synthetic mixture of known aroma compounds.

The manuscript must be edited by a competent person.  There are many awkward expressions and sentences.  For example  “The application of fire for food preparation is a milestone in human history, bringing in the image of roasting meat for the conclusion of a successful hunt” could be written as “The use of fire in cooking is a significant achievement in human history, as it is associated with the idea of roasting meat as a celebration of a successful hunt, which has become a defining symbol.”

L 10.  “The development of healthier and more sustainable food products, such as plant-based meat substitutes (PB-MS), has received a lot of interest and resources in recent years. A thorough understanding of the aroma composition can support efforts to improve the sensory properties of these products and promote their acceptability by consumers.”

The abstract need not provide justification of the purpose of the study.  It should provide the most important discoveries reported in the paper.   Suggest the abstract should be rewritten.  The abstract must indicate how it was done rather than what was done.

L 70,  delete “seamlessly”

L71.  “Electron-impact (EI)” is not recommended anymore.  Use electron-ionization MS.

L 104 “[Error! Bookmark not defined.],”  ??

L 105.  “an injection port” There are two injection ports and two columns in Figure 1.  The purpose and the advantage of having two different columns is not very well explained.  Only one system is used at a time?

L 109.  The purpose and the advantage of having two different columns must explained.  Only one system is used at a time?  Two samples are injected simultaneously?

L 110.  Interchangeable? Meaning?  Use proper technical language.

L 111. “Each inlet is connected separately to a capillary column, and the gas flows are combined using two ports of a zero dead-volume using three-way splitter with makeup gas, to control the total gas flow independently of the column flows  The purpose of this must explained.  Are both columns identical in size and properties? 

L 114. “from a third port”  Is this not shown in Figure 1?  Overall gas splitting descriptions are very difficult to follow.

L 119 “splitted” ?? split

L 117.  “a dead volume 3-port splitter” what is a “a dead volume 3-port splitter”?

L 118.  The use of multiway splitters is difficult to follow.

L 125. “hence the data integration is particularly useful  Tell the readers how this is useful.

L129.  If I understand correctly, the second column simply acts as a make-up gas line.  In reality, a stationary phase coated chromatographic column is not necessary. 

L131. “for implementation of complementary methods” Explain.

L134.  “For the current study, an auto sampler was installed at the back inlet for injection of standards and calibration mixtures, while SPME injection was performed at the front MMI inlet.”  Move this to the beginning of the paragraph.   

L 136. Same data folder.

L 182.  Figure 2. Algorithm of AromaMS software tool - following TIC deconvolution with AMDIS and 183 identifications in with or without verification with selective detection for nitrogen and sulfur” Figure 2 does not show an algorithm.

L 185-204.  The usefulness of all these sentences to a reader is highly questionable.  Unless some specific examples and data obtained are described.

L 221.  Most pyrazines have not been exactly identified.  If method cannot identify the correct isomer, this analytical procedure does not hold much promise.   

L 229. Pyrrole and dimethyldisulfide peaks are only 229 partially resolved

L 231. “The four compounds in the center elute in less than 8 seconds”  Rewrite properly.

L 239.  Figure 3. Dimethylpyrazine, ethyl,methylpyrazine, ethyl,dimethylpyrazine peaks must be properly odentifies.  Or tell that this procedure in unable to do it.

L 239.  Figure 3. What is the huge peak at 0.854 in NPD trace.  Dimethylmethylamine.  How was this tailing peak integrated for Table S4 data?

L 273.  dimethyl pyrazine.  All compounds must be properly named. dimethyl pyrazine should have no space.   Most importantly, this study is mainly about characterizations of aroma compounds without any ambiguity.  There are three isomeric dimethyl pyrazines.    

L 282. Label with arrows the ethyl,methylpyrazine, and trimethylpyrazine peaks.   In Panel C, what are the peaks in 0.87-0.91 from three different samples; why retention times are different?  Describe what is in the inset?

L 293 “The uniformity of the GC-MS analysis is a drawback for the interpretation of the results?  What is the uniformity?  Why it is a drawback?

L 305.  “split into three detectors has been shown to 305 be robust to operation and accurate for detector time-line correlation.”  Rewrite.

L 313.  Compounds or samples?

L 321.  Define the samples in more detail,  ??% lean meat etc.   PB-MS give the brands etc.

L 326. w/65um  ?  micrometer?

L 327. “7330-Urespectively” ?

L 333.  15 m x 0.25 mm x 1 mm

L 334.  What make up gas? 

L 334. “makeup gas ( P/N G2855- 334 61500 by Agilent)”  Agilent supplies makeup gas?

L 352. “with vent time of 0.5 min column”  what is the meaning of this “column”?

L 357. “according to vendor recommendations” what were the recommendations?

L359.  Can the authors provide the code as a supplementary file. If the program is not downloadable or commercially available, it is not useful to the readers of this paper.

L 372. “efficient characterization of N/S-containing compounds in a complex mixture.”  But pyrazines were not properly characterized?

Supplementary data

1)      Should include a Table contents

2)      Tabl1 S1. The column heads must include units.  RT should be minutes etc.

3)      The discrepancies between the experimental RI and NIST values must be explained.   They are often very significantly different.    The same stationary phase has been used?  In any case the data as given does not provided any confidence for a reader to use this method.

4)      (3S)-(-)-3-Acetamidopyrrolidine.  It is not possible for this method to provide the absolute configuration.  This is simply a cut and paste from the NIST data.

5)      The boron compounds identified, are they really found in aroma!  Well are they simply contaminants or analytical artifacts?

6)      4,4'-Bitriazolyl  ??  Is an incomplete name. 4,4'-Bi(1,2,4-triazole)

Author Response

Dear reviewer,

Please find attached our revised manuscript molecules-2317462, “Aroma characterization of roasted meat and meat substitute using gas-chromatography mass-spectrometry (GC-MS) with simultaneous selective detection and the dedicated software tool AromaMS”, submitted for publication in Molecules. We were happy to learn that the topic is of interest to the scientific community.

In particular, we appreciate the thorough and detailed proofreading, and have submitted the manuscript for English language editing by Editage Author service to improve its level of English writing.

We hope that you will now find the attached revised manuscript suitable for publication in Molecules.

Sincerely,

Nitzan Tzanani, PhD

Analytical Chemistry Department,

Israel Institute for Biological Research (IIBR)

Round 2

Reviewer 3 Report

L 266.  “uncertainly identified” ?  “ tentatively identified.

Figure 6 and elsewhere it should be stipulated, that the exact identity of compounds such as dimethylpyrazine was not determined.

Author Response

I would like to thank the reviewer for his comments

The comments are appreciated and the manuscript was corrected accordingly, emphasizing that all identifications are based on library search, hence should be referred to as tentative.